# PIN: Prolate Spheroidal Wave Function-based Implicit Neural Representations

**Dhananjaya Jayasundara**[1*], **Heng Zhao**[2,3], **Demetrio Labate**[2], **Vishal M. Patel**[1]
[1]Johns Hopkins University, [2]University of Houston, [3]Data Science Platform, The Rockefeller University
{vjayasu1,vpatel36}@jhu.edu,{hzhao25@central, dlabate@}.uh.edu

## ABSTRACT

Implicit Neural Representations (INRs) provide a continuous mapping between the coordinates of a signal and the corresponding values. As the performance of INRs heavily depends on the choice of nonlinear-activation functions, there has been a significant focus on encoding explicit signals within INRs using diverse activation functions. Despite recent advancements, existing INRs often encounter significant challenges, particularly at fine scales where they often introduce noise-like artifacts over smoother areas compromising the quality of the output. Moreover, they frequently struggle to generalize to unseen coordinates. These drawbacks highlight a critical area for further research and development to enhance the robustness and applicability of INRs across diverse scenarios. To address this challenge, we introduce the **P**rolate Spheroidal Wave Function-based **I**mplicit **N**eural Representations (PIN), which exploits the optimal space-frequency domain concentration of Prolate Spheroidal Wave Functions (PSWFs) as the nonlinear mechanism in INRs. Our experimental results reveal that PIN excels not only in representing images and 3D shapes but also significantly outperforms existing methods in various vision tasks that require INR generalization, including image inpainting, novel view synthesis, edge detection, and image denoising.

## 1 INTRODUCTION

Discrete representations of images and shapes currently dominate computer vision tasks, with benefits derived from their convenience in storage and computation. Nonetheless, it is worth noting that memory consumption escalates exponentially with the dimensions and resolutions of the data. Moreover, the discrete formation limits the expressivity with the finite discrete grid size or volume size. Implicit Neural Representation (INR) (Sitzmann et al., 2020; Tancik et al., 2020) provides an alternative mechanism that continuously represents signals parameterized through Multi-layer Perceptions (MLPs). INRs are usually fully connected neural networks designed to learn a continuous mapping between coordinates of a signal to the corresponding signal values, resulting in a compact representation of the signal with arbitrary resolution. Beyond representation (Chen et al., 2023b; Saragadam et al., 2022), INRs have demonstrated prominent advantages in various vision tasks, including image reconstruction (Czerkawski et al., 2021; Saragadam et al., 2023), medical imaging (Shen et al., 2022; Sun et al., 2021), and novel view synthesis (Barron et al., 2021; Mildenhall et al., 2020; Niemeyer et al., 2020).

Despite the evident advantages delineated above for INRs in vision computing tasks, properties of some inherent structures of INRs present obstacles to a more widespread application in various vision tasks. First, the expressivity of INRs is heavily influenced by the choice of activation function (Sitzmann et al., 2020; Ramasinghe & Lucey, 2022; Saragadam et al., 2023). The original INRs employing ReLU as activation function were demonstrated to exhibit poor performance in signal representation tasks due to the spectral bias of MLPs (Basri et al., 2020; Rahaman et al., 2019), which favors learning low-frequency and lacks the ability to represent fine details. Also, when reconstructing from sparse measurements, INRs tend to overfit to the coordinates used during training, i.e., the INR can only accurately represent the signal values of the trained coordinates, leading to blurry or noisy reconstructions.

---

*Corresponding Author

Fourier features (Tancik et al., 2020) or positional encoding (Mildenhall et al., 2020) is a transformation which maps low-dimensional coordinates to high-dimensional features, enabling MLP's input layer to embed high frequencies and learning the high-frequency content of a signal (Tancik et al., 2020). Several alternative nonlinearities including sinusoids (Sitzmann et al., 2020), Gaussian functions (Ramasinghe & Lucey, 2022), and Gabor Wavelets (Saragadam et al., 2023) have been proposed as replacements for ReLUs along with positional embedding schemes, often leading to notable enhancements in signal encoding capabilities. One key property of Gaussians and Gabor Wavelets is their good balance of joint space-frequency energy concentration, which explains their efficient performance according to classical signal processing. Nevertheless, another limitation of current INR design is their high sensitivity to hyperparameter selection of parameters. Gaussian and Gabor wavelet nonlinearities require to set a predefined frequency or scale parameter for the INR to achieve competitive performance. How to determine best hyperparameters for the INR activation function and how to initialize the network is still heuristic, and highly dependent on the processing task.

Numerous recent studies have explored the inherent properties of INRs through theoretical analyses and experimental investigations to advance the understand of the mechanisms behind their success and limitations. The Neural Tangent Kernel (NTK) (Jacot et al., 2018) was used to reveal that standard MLP converges very slowly to high-frequency signals in low-dimensional coordinate-based INRs, owing to the rapid frequency fall-off of its corresponding kernels.When utilizing Fourier features (Tancik et al., 2020) or sinusoidal activation functions (Sitzmann et al., 2020) or their recently introduced variants (Liu et al., 2024; Shi et al., 2024; Kazerouni et al., 2024), which aim to achieve superior representation through variable periodic activation functions, sinusoid adjustments based on deep prior knowledge, and Fourier reparameterization, respectively, the Neural Tangent Kernel (NTK) is transformed into a stationary (shift-invariant) kernel, enabling control over the range of learned frequencies (Tancik et al., 2020). The work by Yüce et al. (2022) provided a theoretical analysis of the expressivity and inductive bias of INRs, as well as the imperfect recovery resulting from the inadequacy of input frequencies to properly capture the frequency of the signal. Experimentally, Saragadam et al. (2023) investigated the first layer output of INR, demonstrating that the spatial and frequency compactness of the activation function, as well as the multi-dimensional non-linearity, provide more accurate representations for natural images. Roddenberry et al. (2023) demonstrated that the output of an INR with a generic wavelet activation in the first layer can be expressed in terms of the same wavelets.

In this paper, motivated by these recent theoretical and experimental findings, we propose the employment of a novel activation function for INRs: the Prolate Spheroidal Wave Function (PSWF). PSWFs are designed to maximize the spatial and frequency domain energy concentration, and have previously been used in many vision applications (Lindquist & Wager, 2008; Wendt et al., 2010; Brown & DC, 1968) demonstrating that having an optimal balance of energy concentration in both spatial and frequency domains is critical for efficient signal representation and approximation (Ramasinghe & Lucey, 2022; Saragadam et al., 2023). Our results below showcase that emplying PSWFs as the activation function for INRs endows this architecture with greater expressivity and generalizability as compared to sinusoidal, Gaussian, and Gabor functions which are currently used in state-of-the-art INR implementations. *We attribute this improved performance to the high energy compactification of PSWF atoms and their flexibility, namely the ability to easily tune their hyperparameters to various tasks*. Extensive numerical experiments demonstrate that our new INR model excels not only in representation tasks but also in more challenging reconstruction tasks, e.g., image inpainting, where exisitng INRs often perform poorly.

## 2 RELATED WORKS

**Implicit Neural Representation.** INRs have emerged in recent years showcasing remarkable performances not only in signal representation tasks but also in many inverse vision applications, e.g., unordered signal representation such as 3D shape representation (Park et al., 2019; Mescheder et al., 2019) or novel-view synthesis (Barron et al., 2021; Mildenhall et al., 2020; Niemeyer et al., 2020). The INR represents signals by parameterizing the mappings between input coordinates and signal values using an MLP. The MLP usage here differs from the standard approach in that the inputs are low-dimensional coordinates instead of high-dimensional pixels. In such cases, the ReLU activation function performs poorly due to its *spectral bias*. Recent works have improved the expressivity

of INRs by incorporating coordinate transformation or introducing a special type of non-linearity for the MLP, resulting in enhanced performance in signal representation tasks. Unlike conventional neural networks, the training of INRs is always conducted on a case-by-case basis, which hinders their widespread adoption in vision computing tasks. Several works have proposed accelerating training through multi-scale methods (Saragadam et al., 2022), local blocks (Reiser et al., 2021), Mixture-of-Expert (Wang et al., 2022), adaptive coordinate (Martel et al., 2021).

**Expressivity of INRs.** Despite the widespread adoption of INRs in various tasks, a thorough theoretical understanding is still limited. Fourier analysis was used to explore the *spectral bias* of standard MLPs with ReLU (Rahaman et al., 2019; Cao et al., 2019; Basri et al., 2020), i.e. low frequency is learning fast and robust to noise. The use of Fourier Feature Network (FFN) (Tancik et al., 2020) or periodic non-linearity (SIREN) (Sitzmann et al., 2020) can alleviate spectral bias by transforming the NTK to a stationary (shift-invariant) kernel. Building upon the basic property of trigonometric function, Yüce et al. (2022) demonstrated that the expressivity of FFN and SIREN essentially share the same expressive power, which is characterized by the linear combination of sinusoidal functions at integer hamonics. They also identified the reason for the failure recovery as resulting from the uncovered spectral components, which leads to severe artifacts. For non-periodic activations, such as wavelet (Saragadam et al., 2023),Roddenberry et al. (2023) derived a bound of functions that can be represented by INR.

## 3 METHODOLOGY

### 3.1 FORMULATION OF AN INR

An INR encodes the mapping between input coordinates $\mathbf{r} \in \mathbb{R}^I$ and corresponding signal values $f(\mathbf{r}) \in \mathbb{R}^O$, denoted as $g : \mathbb{R}^I \to \mathbb{R}^O$, and the mapping ,which is $g$, is parameterized by a fully connected neural network $\Phi_\theta : \mathbb{R}^I \to \mathbb{R}^O$, where $\theta$ represents the parameters of the neural network. For instance, when representing an image, the INR maps pixel coordinates $\mathbf{r} = (x_i, y_i)$ to the corresponding RGB values $f(\mathbf{r}) = (R_i, G_i, B_i)$. The fully connected network $\Phi_\theta$ typically an $L$ layer MLP, each layer given by:

$$
\begin{aligned}
\boldsymbol{z}_0 &= \gamma(\boldsymbol{r}) \\
\boldsymbol{z}_k &= \sigma(\boldsymbol{W}_k \boldsymbol{z}_{k-1} + \boldsymbol{b}_k)), k = 1, \ldots, L-1 \\
\boldsymbol{z}_L &= \boldsymbol{W}_L \boldsymbol{z}_{L-1} + \boldsymbol{b}_L.
\end{aligned}
\tag{1}
$$

where $\gamma$ is the position encoding, $\sigma$ is the nonlinear activation function. $\boldsymbol{W}_k$, $\boldsymbol{b}_k$ are weights and biases of the $k$'th layer. $\boldsymbol{z}_0 \in \mathbb{R}^I$ is the input coordinate and $\boldsymbol{z}_L \in \mathbb{R}^O$ is the output of final layer.

### 3.2 WHY PSWFS?

In the recent INR literature, periodic functions such as the sinusoidal activation function and its analogous Fourier embedding have shown to yield better approximations than ReLU, particularly for reconstructing high-frequency components. Functions with better spatial localization, such as Gaussian (Ramasinghe & Lucey, 2022) or Gabor wavelets (Saragadam et al., 2022) have shown higher accuracy in image and shape representation tasks, benefiting from their analytic properties. Even though INRs with these activations work competitively in image and shape representation problems, they work poorly for reconstruction tasks from sparse or noisy measurements, such as image inpainting and denoising. Furthermore, when encoding the signals, existing INRs have a tendency to lose the balance between smoother and finer detailed areas in favor of finer details. A reason for these limitations could be the inadequate energy compactification of INRs when employing Gabor wavelets or Gaussians. This problem is demonstrated by the great sensitivity of INRs to the selection of scale and frequency parameters for Gaussians and Gabor wavelets (Saragadam et al., 2023). Signal representation can be improved by adjusting these parameters for each signal to make such activations more compact in space. However, fine tuning those parameters from signal to signal is extremely tardy, and inefficient task. In addition to that, when INRs are trained on partial data and evaluated on the entire dataset, they fail to generalize, even with different parameter settings. The recovered-masked areas tend to be purely noisy. Similarly, in image denoising, the INRs tend to learn the noise, making it difficult to remove effectively. These observations motivate us to explore alternate activation functions with better space-frequency compactification and lead to PSWFs. Figure 1 shows the space-frequency trade-off of the existing activations along with PSWF.

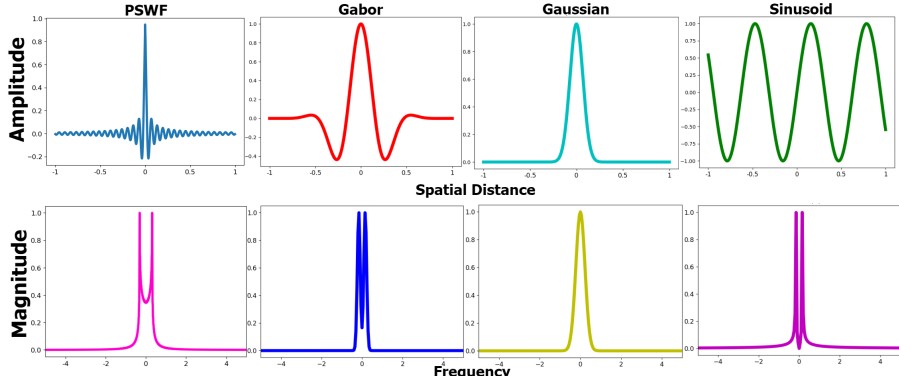

Figure 1: **Space-Frequency Tradeoff of Activations**: The top row illustrates how the values of activation functions change with spatial distance, while the bottom row demonstrates how their magnitudes vary in the Fourier domain with frequency. When an activation function compresses in the spatial domain, it tends to have a broader spectrum in the frequency domain, and vice versa. This phenomenon is known as the space-frequency tradeoff of functions. PSWFs are recognized for their optimality in preserving the highest energy across both domains.

### 3.3 Prolate spheroidal wave functions

PSWFs originated from a question posed by Shannon (Moore & Cada, 2004) in the 1960s: *To what extent are functions, which are confined to a finite bandwidth, also concentrated in the time domain?* Slepian (Slepian & Pollak, 1961; Slepian, 1964; Landau & Pollak, 1961b; 1962), discovered that such functions are associated with the solutions to a Sturm-Liouville problem arising from the Helmholz equation on the prolate sphere (hence the name). In a seminal sequence of papers, Slepian, Pollak and Landau at Bell Labs introduced an integral formulation for this energy concentration problem which led to the following equation (Landau & Pollak, 1961a; Slepian & Pollak, 1961) where the PSWFs $\psi_n(c, t)$ are the eigenfunction solution of the integral operator problem

$$\int_{-t_0}^{t_0} \psi_n(c, t) \frac{\sin \Omega(x - t)}{\pi(x - t)} \mathrm{d}t = \psi_n(c, x)\lambda_n(c),$$

where $\lambda_n(c)$ are corresponding eigenvalues, $c$ is the bandwidth parameter and $c = t_0\Omega$, $\Omega$ is the cut-off frequency of $\lambda_n(c)$. The PSWFs form an orthonormal basis of the space of $\Omega$-bandlimited functions with the fundamental property of being maximally concentrated in time and frequency domain. This energy concentration property has been proven to very advantageous in various signal processing tasks (Gosse, 2013; Hu et al., 2014; Lindquist & Wager, 2008), most notably in sampling problems where PSWFs have been employed for image reconstruction from sparse samples (Khare & George, 2003; Hogan et al., 2010; Lindquist, 2003). The success of PSWFs in sampling applications is of particular interest here, as it suggests their potential for efficiently representing complex information. In our results section below, we confirm that leveraging excellent expressivity properties with robustness in the sparse and possibly noisy sampling setting; PSWFs leads to a remarkably effective choice of activation function for INRs.

In an INR model that employs polynomial activation functions and compactly supported wavelets as template functions in the first layer, Roddenberry et al. (2023) demonstrates that the Fourier transform of the INR output can be characterized by the convolutions of the Fourier transforms of the first-layer atoms with themselves. This observation can be extended to PSWFs which have not compact support but have rapid space decay. They are a generalization of Legendre polynomials and can be represented by the expansion:

$$\psi_n(c, t) = \sum_{k=0}^{\infty} \beta_k^n P_k(t), \tag{2}$$

where $P_k(t)$ is normalized version of Legendre polynomial of order $k$ where the coefficients $\beta_k$ can be calculated by recurrence relation as shown in Moore & Cada (2004).

As shown in Roddenberry et al. (2023) and Yüce et al. (2022), the support of scaled and shifted versions of the template in the first layer is preserved, implying that the output at given coordinate relies solely on the template in the first layer whose support contains this coordinate. For PSWFs it is indeed true that the Fourier support is preserved. As a result, the Fourier output at a given coordinate $\xi$ is dependent only upon the PSWFs in the first layer whose Fourier support contains $\xi$.

## 4 SPACE-FREQUENCY LOCALIZATION

Strong localization in both space and Fourier domains is a highly desirable property in classical signal processing which is known to be critical for robust signal approximations and reconstructions (Donoho et al., 1998). In a nutshell, localization entails that the energy of a signal is highly localized, so that complex information can be efficiently decomposed into well localized components. Emerging evidence shows that space-frequency localization of the activation function is equally important in INRs.

For this reason, the selection of PSWFs in the PIN design has a major impact to explain their excellent performnance. The visualization in figure 1 illustrates the behavior of PSWFs across space and Fourier domains. In the spatial domain, we can observe how PSWF is highly localized within a specific region, rapidly tapering off away from the center. This localization is mirrored in the frequency domain, where the function is confined within a finite bandwidth. In section 5 we prove that the output of PIN, $\Phi_\theta(r)$ is a polynomial of PSWF with a degree of $K^{L-1}$. Moreover, the Fourier transform of $\Phi_\theta(r)$ is band-limited and with rapid space decay which contributes to the advantage in INRs.

## 5 LOCALIZATION AND EXPRESSIVITY PROPERTIES OF PIN

Our INR architectures can be decomposed into a mapping function $\gamma(r) : \mathbb{R}^d \to \mathbb{R}^{T_0}$, also called positional encoding, followed by an MLP with weights $\boldsymbol{W}_k \in \mathbb{R}^{T_k \times T_{k-1}}$, biases $\boldsymbol{b}_k \in \mathbb{R}^{T_k}$ and activation function $\rho$ applied element wise. By denoting as $\boldsymbol{z}_k$ the post-activation functions, the INR computes the output function $\Phi_\theta = \boldsymbol{z}_L$ as form equation 1. Our PIN adopts the PSWFs, denoted below with the symbol $\psi$, as the activation function ($\rho = \psi$). We choose $\psi$ to be the PSWF of order 0.

**Theorem 1.** *Let $\Phi_\theta(\mathbf{r}) : \mathbb{R}^d \to \mathbb{R}$ be an $L$ layer PIN, with weights $\boldsymbol{W}_k \in \mathbb{R}^{T_k \times T_{k-1}}$, biases $\boldsymbol{b}_k \in \mathbb{R}^{T_k}$ in $k$-th layer, assume the activation function $\psi$ is PSWF which can be approximated with a polynomial of degree at most $K$, that is $\psi(x) = \sum_{m=0}^{K} \alpha_m x^m$.*

*Then, $\Phi_\theta(\mathbf{r})$ can be expressed as polynomial of $\psi(\boldsymbol{W}_1^{(t)}\gamma(r) + b_t)$ with degree at most $K^{L-1}$*

$$\Phi_{\boldsymbol{\theta}}(\mathbf{r}) = \sum_{m=0}^{K^{L-1}} \sum_{\ell_1+\ell_2+\cdots\ell_n=m} \prod_{t=1}^{n} \hat{\alpha}_{\ell_t} \left( \psi(\boldsymbol{W}_1^{(t)}\gamma(r) + b_t) \right)^{\ell_t}.$$

**Theorem 1** shows that the Fourier transform of $\Phi_\theta(\mathbf{r})$ is a $K^{L-1}$-order convolution of Fourier transforms of PSWFs $\psi$. Since $\psi$ is band-limited, and the convolution of band-limited functions is band-limited, then $\Phi_\theta(\mathbf{r})$ is also band-limited. Additionally, since convolution increases regularity, $\Phi_\theta(\mathbf{r})$ has high-order of regularity in the Fourier domain implying that $\Phi_\theta(\mathbf{r})$ has very rapid decay in space, i.e., it is highly localized.

## 6 ADAPTIVE ACTIVATION FUNCTION PARAMETERS LEARNING

We have argued above that the optimal spatial and frequency concentration of PSWF positively impacts the performance of the INR model. Another critical advantage of this approach is its flexibility to determine the best hyperparameters of the PSWF activation.

As noted above, the parameters of activation functions in WIRE and GAUSS are chosen for different tasks and different signals using a grid search (Saragadam et al., 2023). However, the activation function's parameters resulting from such a grid search are local. i.e., *they are only optimal for the*

*signal that has been used in the grid search*, while they are suboptimal for other signals, affecting the expressivity of the INR model. Even if those parameters are designed to be learnable, the initialization still impacts the parameters of the learned activation function. For instance, the parameters in the Gabor wavelet $\psi(x; \omega, s) = e^{j\omega x} e^{-|sx|^2}$ are $\omega$ and $s$, representing the frequency and scale respectively; in Gaussian function $\psi(x; s) = e^{-(sx)^2}$, the parameter $s$ represents the scale. As the parameters of these functions appear in an exponent, it is difficult to learn wider frequency or scale distributions.

The situation is different for our PSWF activations as we do not use an explicit formulation, but we use a numerical approximation, so there are no explicit parameters to control. Rather, we apply an indirect way to control the amplitude, frequency, and height characteristics of the PSWF. That is, for the predefined PSWF function $\psi(x)$, we instantiate the learnable PWSF activation function by $\hat{\psi}(x) = T\psi(wx) + b$. In this case, the amplitude, frequency, and height of the PSWF activation function are indirectly controlled by $T$, $w$, and $b$, respectively. The construction of PSWFs as an activation, detailed explanations and ablations on adaptive activation function parameter learning are provided in section A.1 in Appendix.

# 7 Experiments

## 7.1 Image representation

Image representation, commonly referred to as image regression, is a measure that evaluates the expressive capabilities of INRs. A superior INR should not only represent the original signal but also preserve a substantial amount of the structural information, texture, color, and contrast of the image. For assessing the effectiveness of INRs in image representation tasks, the Kodak Lossless True Color Image Dataset (Franzen, 1999) was employed. This dataset was chosen to analyze the performance of the PIN across a comprehensive set of images rather than limiting the evaluation to a single example. Comprising 24 images with varied spatial and frequency content, the dataset presents a robust challenge, requiring the PIN to maintain high fidelity in representation. PIN was subsequently trained on each image in the dataset. The left panel of figure 2 showcases the 15[th] image from the dataset alongside the image representations generated by various INRs. On the right of the same figure displays the variation in PSNR across the dataset for different INRs, where each image is indexed along the circumference of the radar plot. The PSNR values for PIN are highlighted in black.

The child image on the left of figure 2 showcases the PIN's color accuracy and its capability of extracting highly varying contents of the image into the weights and biases of an INR compared to existing INRs. A closer look at the decoded representations of the existing baselines reveals that INRs which are only frequency-compact, like SIREN, tend towards a low-pass representation of the explicit representation, failing to capture intricate details. On the other hand, INRs that are both space-frequency compact but with rapid decay in both domains showcase the ability to capture fast-varying components of images into the INR. However, when they focus more on finer details, they fail to balance smooth regions, introducing some additional noise. As PSWFs are the optimal representation in both domains, PIN focuses on both finer and smooth regions, encoding the signal optimally with minimal distortion. Due to PIN's ability to effectively focus on both high and low-frequency components, it outperforms the existing baselines across all images in the dataset, consistently delivering a PSNR of at least 30 dB. Additional image representation results including a thorough evaluation on DIV2K dataset (Agustsson & Timofte, 2017), and learning curves are provided in the supplementary material.

## 7.2 Solving the wide frequency spectrum challenge via PIN

Even though INRs are proficient at converting explicit signals to implicit representations, their performance is impeded when dealing with signals that encompass a wide frequency spectrum. Specifically, while INRs that incorporate Fourier feature encodings (Landgraf et al., 2022) excel at capturing fine details through high-frequency components, they simultaneously introduce undesirable noisy elements in areas that should remain smooth. Notably, according to the findings in (Landgraf et al., 2022), SIREN (Sitzmann et al., 2020) also exhibits this limitation, struggling to maintain fi-

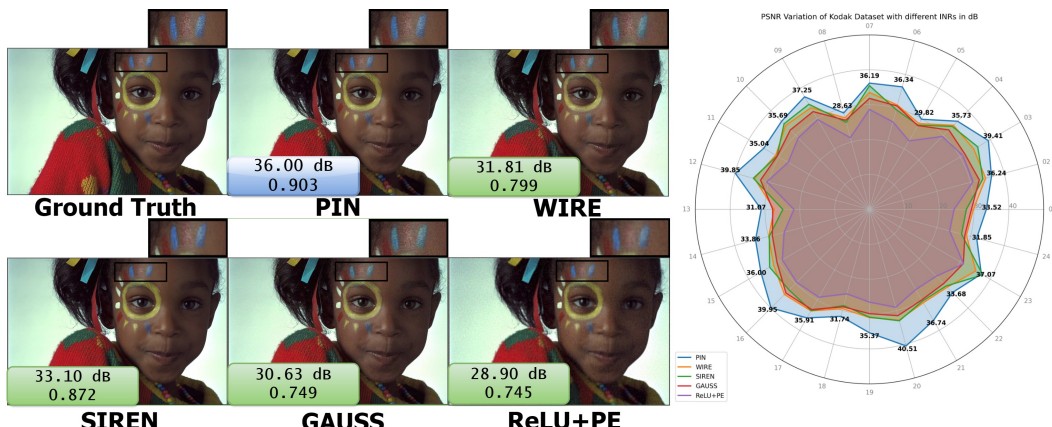

Figure 2: **Image representation capacity of PIN**: The left image shows an instance of PIN's representation capacity, and the right image shows the PIN's representation capacity evaluated on all Kodak lossless true color images. PIN stands out as the INR that achieves the highest PSNR and SSIM metrics, indicating minimal distortion and maximum preservation of structural information.

delity in smoother regions while effectively representing finer textures, and (Landgraf et al., 2022) presents a hierarchical representational approach to resolve this issue. Further, rapid decaying space-frequency non-linearities such as Gabor Wavelet (Saragadam et al., 2022) and GAUSS (Ramasinghe & Lucey, 2022) are good at focusing on fast varying components of images and capturing them efficiently into weights and biases, but when they focus on intricate details, they lose their focus on smooth regions and introduce some form of noisy component (see figure 3). However, as PSWFs are known for being optimally space and band-limited, they excel in the preservation of energy concentration. This characteristic allows them to adeptly represent finer details without encountering this issue. Therefore, PIN achieves an optimal representation that maintains high fidelity across different textural features. This effect is illustrated in figure 3, where the efficacy of PSWFs in balancing detail and smoothness is visually demonstrated compared to the existing INRs. Furthermore, this issue can be seen in the decoded representations of the child image in figure 2 as well. Therefore, from the presented visual results, it can be clearly seen that without the need for any additional architectural changes to INRs, PIN can resolve this wide frequency spectrum challenge.

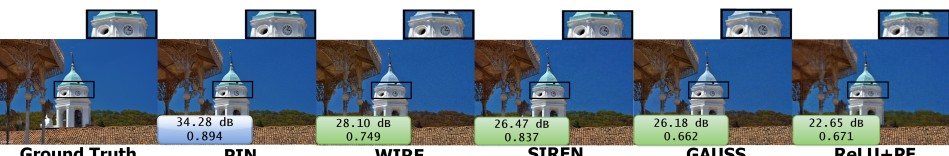

Figure 3: **Wide-Frequency spectrum challenge**: Existing INRs often attempt to emphasize finer details while not focusing much on smoother regions. This can be attributed to their nonlinearities' space-frequency trade-offs. In contrast, PIN exhibits optimal energy concentration for a given band, enabling them to effectively encode both fine details and smooth regions.

## 7.3 OCCUPANCY FIELD REPRESENTATION

When it comes to representing three dimensional occupancy fields, the topic of image representation, as explored in figure 2 extends into the realm of three-dimensional spaces. This extension allows for an examination of the capability of INRs to depict three-dimensional signed distance fields. In this context, the transformation occurs from a three-dimensional domain to a binary signal space, marked by the values 1 or 0. Where, a value of 1 signifies the presence of the signal within a predefined area, whereas a value of 0 denotes its absence from that area. For this experiment two occupancy volumes namely Asian Dragon, and Armadillo, which are shown in figure 4, were obtained from Stanford 3D shape dataset (Stanford University Computer Graphics Laboratory), and sampled on a grid of $512 \times 512 \times 512$, assigning a value of 1 to each voxel inside the volume and a value of 0 to those outside. The methodology for this study was derived from the work mentioned in Saragadam et al.

(2023). The obtained results are shown in figure 4. It can be observed that in both the instances PIN effectively encode the rapidly changing three-dimensional structure into its neural representation. In contrast, when examining the performance of WIRE and SIREN, it is evident that these architectures struggle to incorporate the quickly varying components of the structure into their weights and biases, resulting in a representation that primarily captures lower frequency components. While GAUSS activation function achieves performance metrics similar to PIN, a detailed examination of the three-dimensional statues decoded from Gauss reveals distortions in some uniform areas in both instances. Conversely, PIN encodes the structure accurately without introducing any additional artifacts.

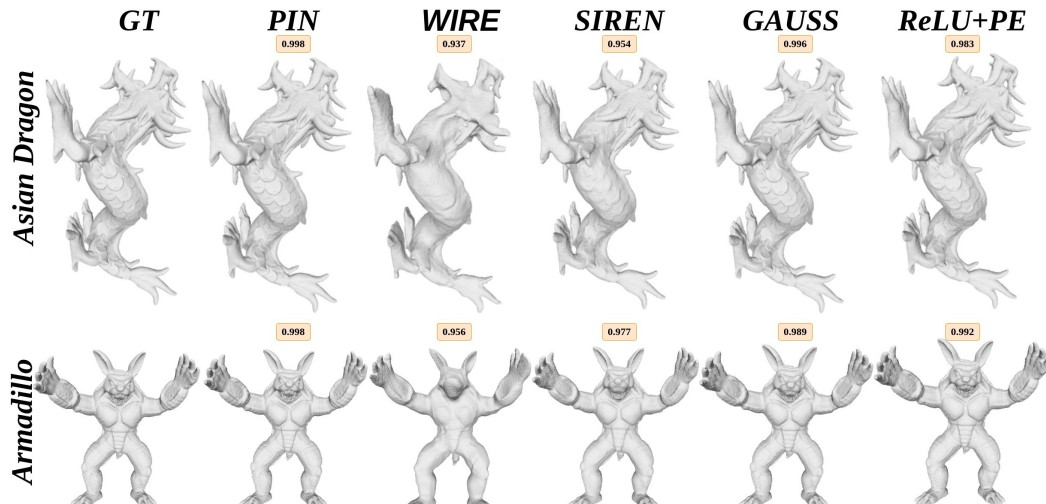

Figure 4: **Occupancy fields representation capacity of PIN**: PIN stands out as the INR that exhibits the highest similarity to the ground truth occupancy field. Unlike other INRs, the PIN approach does not converge to a low-pass representation; rather, it accurately preserves high-frequency components throughout the training process. This characteristic enables PIN to faithfully capture intricate details, making it particularly adept for tasks requiring accurate representation of complex occupancy patterns.

## 7.4 IMAGE INPAINTING

INRs are often employed to learn continuous functions from discretized signals. Once an INR is adequately trained, it should demonstrate excellent generalization capabilities even with limited training samples. A notable application in computer vision is image inpainting. In this approach, during the training phase, an INR receives partial data from an image. In the testing phase, the INR is tasked with reconstructing the missing positions of the image. Image inpainting is a measure to determine if the learned underlying representation has overfitted to the provided training data.

We employ two testing strategies: one involves training with 70% of the data sampled randomly, and the other uses a predefined text mask that obscures the image with varying font sizes. The top row of the figure 5 shows inpainting results, corresponding to 70% random sampling, whereas the bottom of the same figure shows the text-masked inapinting results. In both instances the second column represents the masked image. As can be clearly seen from the results, PIN is the only architecture that maintains the highest PSNR value in both instances, indicating superior recovery capabilities with minimal distortion to the original image. Additionally, it is the architecture that produced the most visually appealing reconstruction in comparison with the ground truth.

When comparing the performance of various INRs, it is evident that WIRE exhibits signs of overfitting to the training data and has not generalized well. SIREN demonstrates commendable generalization capabilities within the context of INRs, yet it is constrained by its need for highly specific weight initialization and frequency parameters. Therefore, PIN is particularly effective for image inpainting, producing visually appealing results even when reconstructing highly obscured images. This underscores its robust generalization ability compared to existing compactly supported INRs like WIRE and GAUSS, and sets a new benchmark that surpasses current state-of-the-art methods.

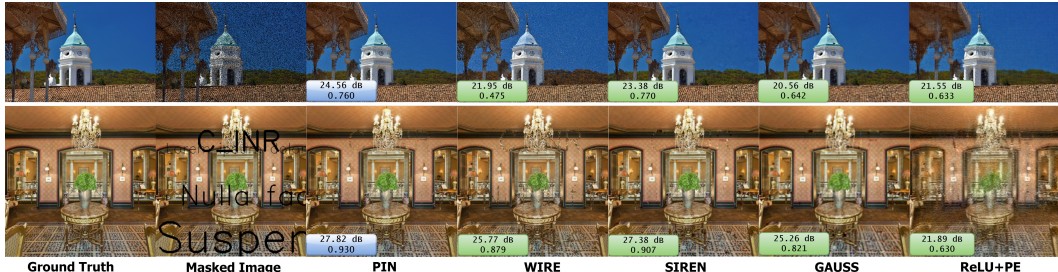

Figure 5: **Image Inpainting Capabilities of PIN**: Among the existing INRs, PIN stands out as the top image inpainting performer, excelling not only in achieving the highest PSNR in both instances, but also in generating the inpainting results most visually similar to the ground truth. PIN's superior metrics highlight its proficiency in restoring images with minimal distortion while preserving maximum structural information.

### 7.5 NEURAL RADIANCE FIELDS

The core area benefiting from INRs is NeRFs (Mildenhall et al., 2020). In NeRF, a scene is encoded into an INR by feeding the viewer's coordinates $(x, y, z)$ and the viewing directions $(\theta, \phi)$ into the INR. The INR is tasked with predicting the corresponding location's color and density. When the INR is well-trained, it gains the ability to generate novel views from different spatial locations and viewing angles, those are not included in the training data. This remarkable capability is achieved by training a 3D implicit function using spatial coordinates $(x, y, z)$ and viewing directions $(\theta, \phi)$ allowing for the synthesis of highly realistic images and scenes. For this experiment, we used a vanilla NeRF architecture consisting of two fully connected blocks, each containing four layers, and the drums dataset with 100 training images and 200 testing images. One of the obtained novel views for each INR is presented in figure 6. PIN's remarkable performance can be attributed to its optimal energy concentration within a specified frequency band, distinguishing it from other methods and reinforcing its superiority. Additional novel views are provided in the supplementary material.

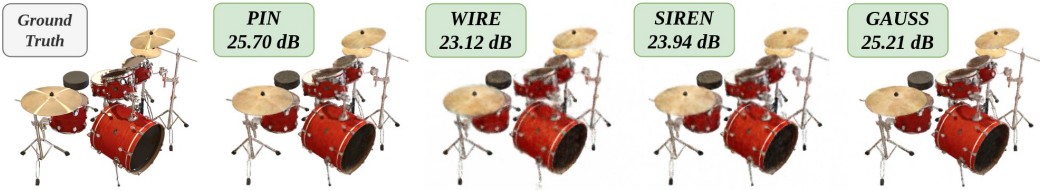

Figure 6: **Novel View Synthesis Capabilities of PIN**: PIN achieves the highest PSNR in novel views performance as compared to existing INRs. While existing INRs tend to produce blurry novel views in feature-dense areas, PIN's ability to preserve maximum energy in a given state allows it to encode intricate details into its weights and biases, and produce novel views that exhibit the highest similarity to the ground truth.

### 7.6 ABLATION STUDY

#### 7.6.1 NETWORK HYPERPARAMETERS

An ablation study was conducted to evaluate the effectiveness of PINs with various hyperparameters selected during the training environment. For this experiment, we used the child image shown in figure 2. This hyperparameter tuning included the variation of PSNR with the number of hidden neurons, while keeping the number of hidden layers constant at 3 (shown in the left figure of figure 7). Additionally, the study examined the variation of PSNR with the number of hidden layers, while maintaining the number of hidden neurons at 300 (shown in the middle figure of figure 7). Lastly, the variation of PSNR with the learning rate was analyzed, with the number of hidden layers kept at 3 and the number of hidden neurons at 300 (shown in the right figure of figure 7)

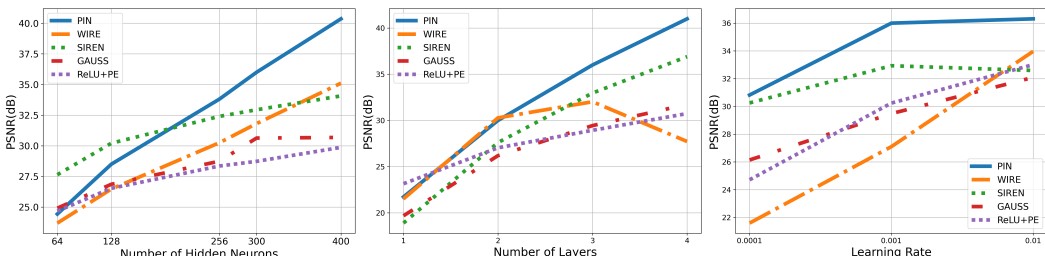

Figure 7: **Hyperparameter Turning of PIN**: PIN demonstrates a sharp linear increase in PSNR with the addition of more hidden neurons and layers compared to existing INRs. Instead of becoming unstable with higher learning rates, PIN stabilizes its PSNR, maintaining nearly constant performance.

In general, increasing the number of hidden neurons or hidden layers in an INR enhances its capacity due to the higher number of learnable parameters. As shown in the left and middle plots of figure 7, PIN exhibits an approximately linear increase, with a higher gradient compared to other INRs, in PSNR with the addition of more hidden neurons and hidden layers. However, as the learning rate increases, PIN demonstrates PSNR saturation effects instead of tending toward instability, as illustrated in the right plot of figure 7.

## 7.7 THEORETICAL ANALYSIS, ABLATION STUDIES, AND ADDITIONAL EXPERIMENTS

In the appendix, we provide a comprehensive theoretical analysis of PIN's forward propagation, alongside a combined theoretical and experimental investigation into why existing baseline space-frequency compact INRs fall short compared to PIN. This analysis helps to elucidate the superior performance of PIN in capturing complex details. The appendix also includes the complete numerical implementation of PIN, where we demonstrate its robustness to variations in parameters and provide a detailed examination of how different weight initialization strategies impact its performance. Furthermore, we showcase PIN's learning curves and extensive experimental results to support the findings presented in the main sections. In addition to these results, the appendix features further analysis of PIN's capabilities through a set of additional experiments. These experiments explore PIN's ability to handle high-frequency encoding, perform effective image denoising, and detect edges with precision.

## 8 CONCLUSION

Implicit Neural Representations (INRs) have emerged as a very promising framework in computer vision and image processing, yet their performance is very sensitive on the choice of activation functions. Empirical results indicate that well-localized activation functions with good space-frequency concentration are more effective than sinusoidal functions which suffer from poor spatial decay. For instance, current INRs often focus more on encoding intricate details into weights and biases of the INR, and do not handle well smoother regions. Further, they underperform in reconstructing sparse or noisy data due to overfitting. To overcome these limitations, we propose the use of PSWF as activation function for INRs. PSWFs maximize joint space-frequency domain concentration, a crucial feature for enhancing INR capabilities. Our numerical experiments demonstrate that INRs with PSWF activations significantly improve expressivity and generalizability, boosting performance across various tasks including image and 3D shape representation, novel view synthesis, reconstruction from sparse or noisy measurements, and edge detection.

## 9 ACKNOWLEDGMENT

This work is supported by the Intelligence Advanced Research Projects Activity (IARPA) via Department of Interior/ Interior Business Center (DOI/IBC) contract number 140D0423C0076. The U.S. Government is authorized to reproduce and distribute reprints for Governmental purposes notwithstanding any copyright annotation thereon. Disclaimer: The views and conclusions contained herein are those of the authors and should not be interpreted as necessarily representing the official policies or endorsements, either expressed or implied, of IARPA, DOI/IBC, or the U.S. Government.

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

# A APPENDIX

## A.1 ANALYSIS

### A.1.1 THEORETICAL ANALYSIS

Let us examine the forward propagation of the PIN.

First, we use positional encoding $\gamma(\boldsymbol{r}) : \mathbb{R}^d \to \mathbb{R}^{T_0}$ to map input coordinates to a higher dimensional Fourier embedding. The 1-st layer pre-activation function at the $j$-th node is:

$$\boldsymbol{v}_1^{(j)} = \boldsymbol{W}_1^{(j)} \gamma(\boldsymbol{r})$$

where $\boldsymbol{W}_1 \in \mathbb{R}^{T_1 \times T_0}$ is linear weights, hence, the 1-st layer $j$-th node after PSWF activation function is:

$$\boldsymbol{z}_1^{(j)} = \psi(\boldsymbol{v}_1^{(j)} + \boldsymbol{b}^{(j)}) = \psi(\boldsymbol{W}_1^{(j)} \gamma(\boldsymbol{r}) + \boldsymbol{b}^{(j)})$$

Notice that $\psi$ is bandlimited.

To examine the effect of functional composition in the subsequent layers, we will use the expansion of PSWFs into Legendre polynomials:

$$\psi(x) = \sum_{k=0}^{\infty} \beta_k \, p_k(x),$$

where $p_k$ is the Legendre polynomial of degree $k$. We assume below that the polynomial expansion of $\psi$ is approximated with a polynomial of degree at most $K$:

$$\psi(x) = \sum_{m=0}^{K} \alpha_m x^m$$

The 2-nd layer pre-activation function at the $j$-th node is:

$$\boldsymbol{v}_2^{(j)} = \boldsymbol{W}_2^{(j)} \boldsymbol{z}_1$$

Using the polynomial expansion for the PSWF activation function, we have that the 2-nd layer $j$-th node after PSWF activation function is of the form:

$$
\begin{aligned}
\boldsymbol{z}_2^{(j)} = \psi\left(\boldsymbol{v}_2^{(j)}\right) &= \psi\left(\boldsymbol{W}_2^{(j)} \psi\left(\boldsymbol{v}_1\right) + \boldsymbol{b}^{(j)}\right) \\
&= \sum_{m=0}^{K} \alpha_m \left(\boldsymbol{W}_2^{(j)} \psi\left(\boldsymbol{W}_1 \gamma(\boldsymbol{r}) + \boldsymbol{b}\right) + \boldsymbol{b}^{(j)}\right)^m \\
&= \sum_{m=0}^{K} \alpha_m \left(\sum_{t=0}^{T_1-1} w_{2,j}^{(t)} \psi\left(\boldsymbol{W}_1^{(t)} \gamma(\boldsymbol{r}) + b_t\right) + \boldsymbol{b}^{(j)}\right)^m \\
&= \sum_{m=0}^{K} \sum_{\ell_1+\ell_2+\cdots\ell_n=m} \prod_{j=1}^{n} \tilde{\alpha}_{\ell_j} \left(\psi(\boldsymbol{W}_1^{(t)} \gamma(\boldsymbol{r}) + b_t)\right)^{\ell_j} \\
&= \sum_{m=0}^{K} \sum_{\ell_1+\ell_2+\cdots\ell_n=m} \prod_{j=1}^{n} \tilde{\alpha}_{\ell_j} \left(\psi(\boldsymbol{W}_1^{(t)} \gamma(\boldsymbol{r}) + b_t)\right)^{\ell_j} \quad (1)
\end{aligned}
$$

Eq. 1 shows that the postactivation output of the $j$-th node in the second layer is a polynomial of $\psi\left(\boldsymbol{W}_1^{(j)} \gamma(\boldsymbol{r}) + b_t\right)$ with degree $K$.

Using induction we can get the similar results for the PIN output (this is simlar to Roddenberry et al. (2023)) $\Phi_{\boldsymbol{\theta}}(\mathbf{r}) = \boldsymbol{W}_L \boldsymbol{z}_{L-1} + \boldsymbol{b}_L$ is a polynomial of $\psi\left(\boldsymbol{W}_1^{(j)} \gamma(\boldsymbol{r}) + b_t\right)$ with degree $K^{L-1}$ where $L$ is the number of layers.

### A.1.2 Why does PIN outperform space-frequency compact INRs?

Activations such as Gauss and Gabor are also space-frequency compact, but with rapid decay in both domains. However, as evidenced by PIN's results, PIN consistently outperforms these baselines. The primary reasons for this are its high-energy concentration and the implementation of PSWFs, which can be explained as follows. As PSWFs are derived through an optimization procedure that ensures optimal energy concentration throughout the given space, they must be constructed using numerical methods. Unlike the Gauss and Gabor functions, at any given band, the PSWF has the maximum energy concentration, tapering off with oscillations but not zeroing out all values outside of the support. This optimal decay of PSWFs allows them to retain both high-frequency and low-frequency components, passing essential elements through the activation function without losing critical information due to rapid decay.

For Gauss and Gabor functions, to obtain better results, the parameters $\omega$ or $\sigma$ are often large (e.g., $\omega = 20$, $\sigma = 10$ in the WIRE paper). These parameters make these functions behave like a Dirac delta, zeroing out most values throughout the considered space. We hypothesize that this property makes the parameters of Gauss and Gabor functions inefficiently learnable. This can be explained mathematically and shown experimentally as follows, using the Taylor expansion. For simplicity, consider the expansion of a Gaussian:

$$\exp(-sx^2) = 1 - sx^2 + \frac{(sx^2)^2}{2!} - \frac{(sx^2)^3}{3!} + \dots$$

where $s$ is a trainable parameter. For large $s$, this behaves like a Dirac delta, filtering most data during the forward pass, which may be undesirable for a given signal. Therefore, backpropagation instructs all trainable parameters, including the activation function parameter, to adjust their values to minimize the overall loss. At a certain epoch, backpropagation may seek to expand or compress the activation function to pass more features or limit them, respectively. As these functions are implemented numerically using a truncated Taylor series of order $2n$, the parameter $s$ appears non-linearly in $2(n-1)$ coefficients. Therefore, adjusting $s$ might exhibit a certain level of reluctance, as it must be simultaneously modified at $2n - 1$ places. Consequently, the value of $s$ is unlikely to change significantly, even though backpropagation attempts to adjust it. This means the shape of the activation curve does not change much, preventing optimal filtering of features to be passed to the final layer. This leads to either the loss of valuable information or the passage of redundant information during each forward pass. However, in PIN, we use a discretized solution from the differential equation with cubic spline approximation, denoted as $\phi(x)$, and $\phi(sx)$ is given by,

$$\phi(sx) = a_0 + a_1(sx) + a_2(sx)^2 + a_3(sx)^3$$

Therefore, in PIN, $s$ appears only in three coefficients, making it easier to adjust $s$ in the loss landscape when necessary, leading to quicker convergence. To substantiate this, we conducted experiments with trainable parameters in PIN and WIRE. For this experiment, we perturbed the parameters as described in section A.3.1, applying a variance of 25. The variation of parameters are shown in figure 8, and it illustrates the adjustments of each trainable parameter across epochs for both PIN, and WIRE. In figure 8 the subscript number in each parameter shows to which layer it belongs. For instance, $T_1$ represents the amplitude parameter of PSWF activation in the first hidden layer.

The first three plots of figure 8 show that PIN's parameters quickly adjust in a way that they locate a local minimum, resulting in a PSNR of 35.56 dB. In contrast, the last two plots of figure 8 depict WIRE's slower parameter adjustments due to its reluctance, resulting in a PSNR of 20.40 dB. Without special treatment, these parameters are unlikely to move effectively.

## A.2 Further experimental details

### A.2.1 Experimental setup

PIN, based on PSWFs, concentrates maximum energy in a given frequency band, and outperforms the existing INRs in representation capabilities, accuracy, and generalization across all visual signals. Rigorous experiments show that PIN precisely learns the underlying continuous mapping of INRs, enabling exceptional performance in inverse vision problems. For numerical experiments,

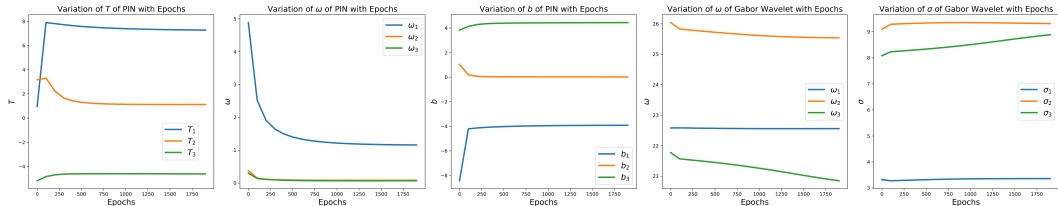

Figure 8: **PIN efficiently locates local minima**: PIN's parameters evolve within a few epochs to a local minimum, resulting in a PSNR of 35.56 dB. In contrast, WIRE's parameters do not move significantly or move very slowly, leading to a PSNR of 20.40 dB. The subscript in each parameter denotes the layer number.

we used the PyTorch framework and the Adam optimizer (learning rate 0.001) with a Multilayer Perceptron (MLP) comprising layers of 300 neurons in each hidden layer. Image performance was assessed using PSNR, SSIM, and IoU for occupancy fields.

### A.2.2 POSITIONAL ENCODING

For the positional encoding, we leverage the method used in (Chen et al., 2023a). Here, we describe the implementation in detail.

Let $W$ and $H$ be the width and height of the image. The coordinate $r_{ij}$, embedded to:

$$\gamma(r_{ij}) = \left[ \frac{i}{2H-1}, \frac{j}{2W-1}, \sin(p_{ij}\boldsymbol{d}_1), \cos(p_{ij}\boldsymbol{d}_1), \cdots, \sin(p_{ij}\boldsymbol{d}_L), \cos(p_{ij}\boldsymbol{d}_L) \right] \tag{3}$$

where $p_{ij} = iH + j$ is the lexicographically ordered location of coordinate $r_{ij}$; $\boldsymbol{d} = e^{\boldsymbol{f}}$, where $\boldsymbol{f}_k = e^{2(k-1)\log(\frac{2l}{L})}$, $k = 1, \cdots, L$, $L$ is the dimension of embedding, $l$ is the length of the sequence.

### A.2.3 NUMERICAL IMPLEMENTATION OF PSWFS

Here, we explain the implementation details of the PSWF activation function. There many methods to sove the integral operator problem:

$$\int_{-t_0}^{t_0} \psi_n(c,t) \frac{\sin \Omega(x-t)}{\pi(x-t)} \mathrm{d}t = \psi_n(c,x)\lambda_n(c);$$

The classical one would be based on the expansion of Legendre polynomial:

$$\psi_n(c,t) = \sum_{k=0}^{\infty} \beta_k^n P_k(t), \tag{4}$$

where $\beta_k^n$ are coefficients of the corresponding Legendre polynomials, which can be obtained by recursions:

$$\underbrace{\frac{(k+2)(k+1)}{(2k+3)\sqrt{(2k+5)(2k+1)}} c^2}_{A_{k,k+2}} \beta_{k+2}^n + \left( \underbrace{k(k+1) + \frac{2k(k+1)-1}{(2k+3)(2k-1)} c^2}_{A_{k,k}} - \chi_n \right) \beta_k^n \\ + \underbrace{\frac{k(k-1)}{(2k-1)\sqrt{(2k-3)(2k+1)}} c^2}_{A_{k+2,k}} \beta_{k-2}^n = 0, \tag{5}$$

recursions equation 5 can be written as:

$$(A - \chi_n \cdot I)(\beta_n) = 0,$$

detailed implementation can be found in Osipov et al. (2013). Here we use $n = 0$, and $c = 1$ to construct our PSWF activation function template $\psi$.

To implement the PSWF as an activation function, we use the prolate spheroidal radial function, then shift and mirror the function according to the $y$ axis to make it suitable for the activation function. The instantiated PSWF function shown in figure 1. Once this discritized mirror reflection is obtained, the next step is to adapt it as a neural network activation function. During forward propagation, the transformed coordinates, computed through linear layers, must pass through the activation function. To obtain this, the activation function needs to be continuous, rather than a set of sampled values. When converting these discrete values into a continuous representation, we employed cubic spline approximation. This method provides a third-order approximation between successive points while ensuring continuity and differentiability. Even though a simple regression could transform the discretized points into a continuous curve, it often fails to retain the exact data points and the differentiability properties between discrete points, both of which are essential for backpropagation. Therefore, cubic spline approximation, which is third-order and computationally efficient, serves as an optimal choice and fulfills the intended purpose.

### A.3 ABLATION STUDIES

#### A.3.1 ROBUSTNESS OF PIN'S PARAMETERS

PIN has three trainable parameters: $T$, $\omega$, and $b$, and they are initialized as $T = 1$, $\omega = 1$, and $b = 0$. To demonstrate the robustness of parameters $T$, $\omega$, and $b$, we conducted an additional four experiments. In these experiments, the parameters were not initialized as $T = 1$, $\omega = 1$, and $b = 0$ but drawn from Normal distributions, denoted as $N$. Specifically, $T$ and $\omega$ are now sampled from $N(1, \sigma^2)$ and $b$ from $N(0, \sigma^2)$, with the first number in the parentheses representing the mean and the second the variance of the distribution. Here we varied $\sigma^2$ and observed the final PNSR. The obtained results are shown in table 1, and each result shown in the table represents the mean of four trials, and the number next to the $\pm$ sign indicates the standard deviation of the obtained four trials. As can be seen from the results, even under a perturbation of variance 25, PIN is able to locate a local minimum in the loss landscape which provides good performance. However, when the variance is 100 for parameters drawn from Normal distribution, PIN only showed a convergence to 1 trial among the 4 trials conducted. Therefore, it can be stated that when PIN's parameters are disturbed by a distribution with a variance of 25, PIN is capable of adjusting its parameters and locating a local minimum. However, when the variance is 100 it is unlikely that PIN converges to a local minimum. To showcase how these parameters are robust compared to baselines, we also experimented with WIRE's parameters under the same perturbations. For WIRE, the optimal values for $\omega$ and $\sigma$ are 20 and 10, respectively. Then, we drew $\omega$ from $N(20, \sigma^2)$ and $\sigma$ from $N(10, \sigma^2)$. The results of these experiments are shown in the table.

In general, PIN's parameters are trainable for all experiments. However, we further analyzed whether the parameters $T$, $\omega$, and $b$ can be used with fixed values. To check this, we trained two different configurations. Config1: $T = 1$, $\omega = 1$, and $b = 0$ for all epochs. Config2: $T$, and $\omega$ are drawn from $N(1, 25)$, and $b$ is drawn from $N(0, 25)$, and the realized values are kept constant. The results are as follows. Config1: PSNR=35.76 dB, and Config2: PSNR=31.66 dB. The training curves for these configurations are shown in figure 9. Therefore, it can be concluded that even when we have a highly perturbed $T$, $\omega$, and $b$ PIN showcases good performance and gets very closer to WIRE. However, in this case due to its inability to move PIN's parameters in the loss landscape such that a way that it finds a better local minimum, it ends up with giving 31.66 dB final PSNR.

Table 1: Robustness of PIN's parameters

| INR | $\sigma^2 = 1$ | $\sigma^2 = 4$ | $\sigma^2 = 25$ | $\sigma^2 = 100$ |
|---|---|---|---|---|
| PIN | 35.86±0.88 | 35.70±0.71 | 35.96±0.75 | Diverges |
| WIRE | 31.69±0.66 | 30.30±1.39 | 20.25±12.24 | Diverges |

#### A.3.2 LEARNING CURVES

Learning curves are a valuable tool that provide insights into how effectively and quickly an INR learns over time. For this task, we selected the 15[th] image from the Kodak dataset, as featured in the

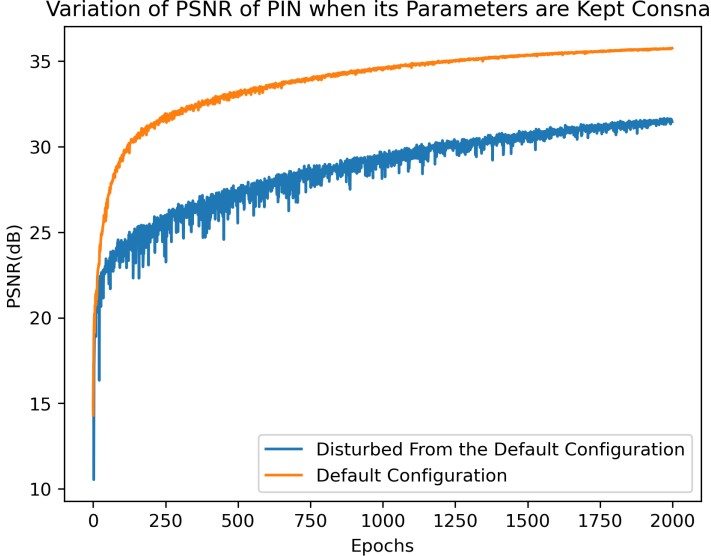

Figure 9: **Variation of PSNR without the explicit control of parameters**: As can be seen, without explicit control of parameters, PIN needs more epochs to obtain convergence. This demonstrates the importance of explicit parameter control for faster convergence.

main paper. The learning curves for various INRs, including PIN, are depicted in figure 10, which illustrates the variations in PSNR (dB) across the number of epochs. Notably, from around the 500[th] epoch, PIN consistently maintains a gap of nearly 3 dB compared to SIREN. This substantial difference indicates that PIN encodes explicit signals into its weights and biases at a much faster rate than any other existing INR.

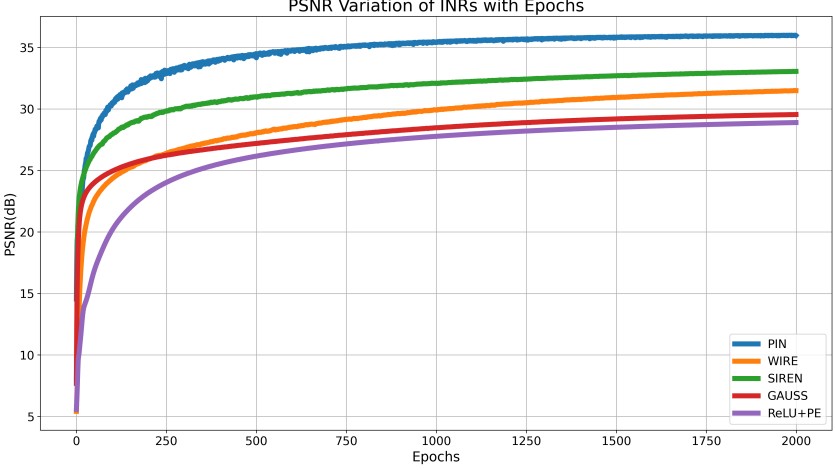

Figure 10: **PSNR variation with the number of epochs**: PIN Learns Faster. The learning curves clearly show that PIN quickly reaches a PSNR of nearly 30 dB within just 125 epochs, and approaches 35 dB by the 500[th] epoch. Consequently, it can be concluded that PSWFs encode explicit representations into an INR both efficiently and rapidly.

### A.3.3 WEIGHT INITIALIZATION

Previous studies (Saragadam et al., 2023; Ramasinghe & Lucey, 2022), have shown that space-frequency compact activation functions like PSWFs are generally insensitive to weight initialization

methods. Consistent with this, all experiments in our paper are obtained using Pytorch's default weight initialization mechanism. However, to experimentally validate this claim, we initialized PIN with different Normal, Uniform, and SIREN-like weight distributions. The resulting PSNRs are: SIREN-like at 35.67, Uniform at 35.39, Normal at 36.06, Pytorch default at 36.00; this shows that weight initialization does not significantly impact the performance.

### A.4 ADDITIONAL RESULTS

#### A.4.1 IMAGE REPRESENTATION ON KODAK

As we have evaluated the capabilities of PIN using the Kodak dataset, some of the decoded representations from PIN, along with other models, are displayed in figure 11. The decoded images are organized into rows corresponding to their positions in the dataset: the first, second, and third rows represent the $1^{st}$, $5^{th}$, and $20^{th}$ images of the Kodak dataset, respectively. In addition to evaluating PIN against WIRE, SIREN, GAUSS, and ReLU+PE, it is also compared against INCODE (Kazerouni et al., 2024) and FR-INR (Shi et al., 2024). The following table summarizes the obtained results across the Kodak dataset.

Table 2: Comparison of methods based on PSNR (dB) on the Kodak dataset.

| Method | PSNR (dB) |
|--------|-----------|
| PIN    | 40.17     |
| INCODE | 34.07     |
| FR-INR | 37.91     |

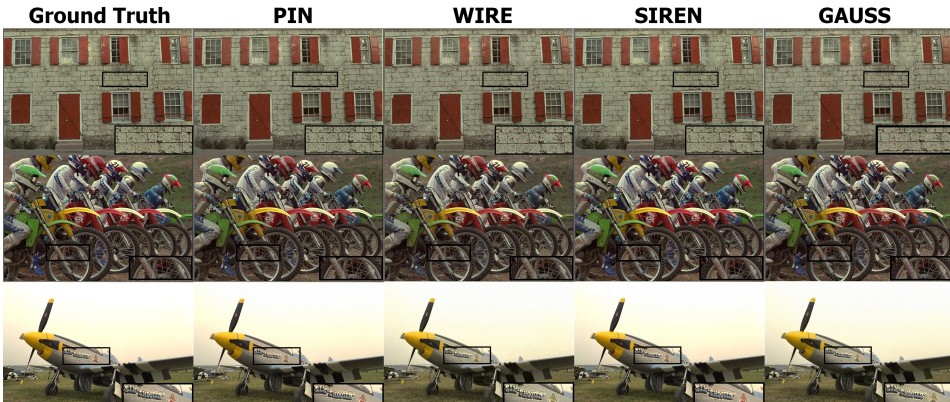

Figure 11: **Additional image representation results**:The decoded representations clearly demonstrate that PIN is the only INR that consistently aligns its outputs with the ground truth. It is evident that some baseline models produce overly smoothed images, and others struggle with accurate color reproduction. Additionally, most baseline models introduce extraneous noise into regions that should be smooth, due to their limited ability to handle signals across a broad frequency spectrum. However, PIN, which is derived from PSWFs, optimally processes each area, thereby producing the most visually coherent decoded images and effectively minimizing distortions.

#### A.4.2 IMAGE REPRESENTATION ON DIV2K

In addition to the evaluation on the entire Kodak dataset (Franzen, 1999), 30 randomly selected images from the DIV2K dataset (Agustsson & Timofte, 2017) were used to demonstrate the effectiveness of the proposed method. These images were represented using different INRs. For this purpose, an MLP with 5 layers and 300 hidden neurons was used. The following table summarizes the obtained results.

Table 3: PSNR Variation on DIV2K Dataset

| Method | PSNR (dB) on DIV2K |
|---|---|
| PIN | 41.46 |
| WIRE | 30.12 |
| SIREN | 38.77 |
| GAUSS | 28.13 |
| ReLU+PE | 26.74 |

### A.4.3  NeRF

Additional views on the novel view synthesis task from various positions and viewing angles are depicted in figure 12. It is evident that the novel views obtained demonstrate PIN's excellence in the task, closely resembling the ground truth. Moreover, intricate details are well-preserved with PIN compared to the other INRs.

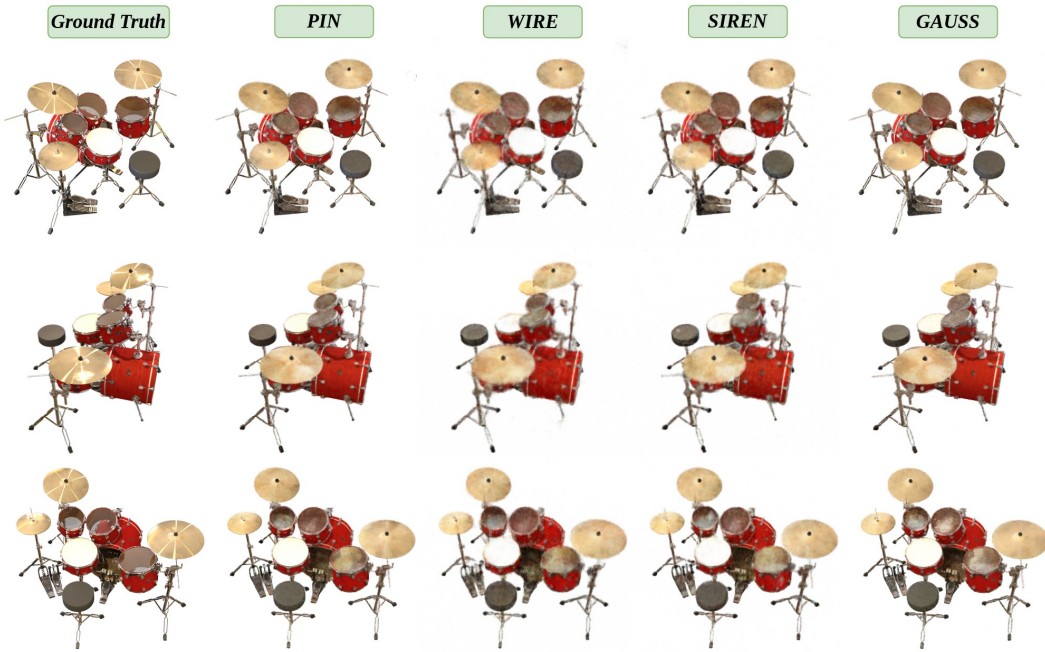

Figure 12: **Additional novel views from different positions and angles**: As can be seen, PIN clearly outperforms other INRs like WIRE, SIREN, and GAUSS, particularly in capturing intricate details of the drum set. The rods supporting the drums are more precisely reconstructed, showing sharper geometry and finer features that are blurred or lost in the other methods. PIN also excels in rendering lighting effects, capturing reflections on the cymbals and shadows across the red drums with greater accuracy. As the synthesized views are closer to the ground truth, PIN stands out as a superior choice for novel view synthesis tasks.

## A.5  ADDITIONAL EXPERIMENTS

### A.5.1  HIGH FREQUENCY ENCODING CAPABILITIES

The ability of INRs to encode high frequencies can be effectively evaluated using images with sharp color transitions. By examining an image featuring abrupt changes in color—corresponding to high-frequency elements in the frequency domain—it is possible to gauge the proficiency of INRs in capturing high-frequency information. For this study, we opted to utilize the image showcased on the left side of the illustration provided in figure 13. Once each INR is trained on this image, the learned

implicit representation is decoded, and the absolute error between the decoded representation and ground truth is obtained. As shown in figure 13, the PIN model yields the lowest residual plot, demonstrating its remarkable signal encoding capabilities. In contrast, INRs such as WIRE and SIREN exhibit inefficiencies not only in encoding sharp transitions but also in handling constant color areas. The exceptional performance of the PIN model may be attributed to the PSWF's optimal management of energy concentration in both spatial and frequency domains

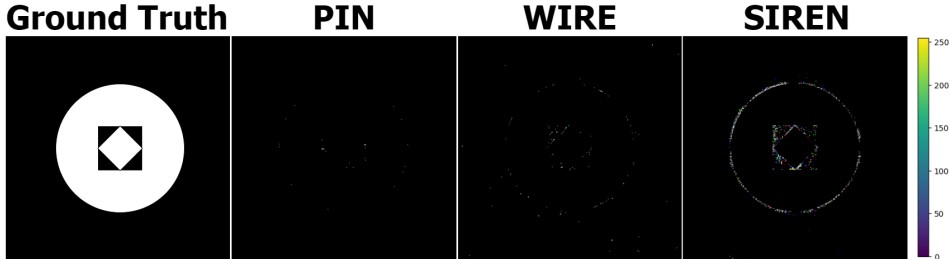

Figure 13: **High frequency encoding capabilities of PIN**: This figure presents the error plots obtained after training each INR on the image shown on the left. It is evident that the PIN produces the smallest residual plot, excelling in encoding high frequencies exceptionally well compared to other INRs, thanks to the optimal energy concentration balance provided by PSWFs. Conversely, WIRE and SIREN exhibit errors not only in encoding high-frequency content but also in constant color areas as well.

### A.5.2 IMAGE DENOSING

In order to find out the possibility of image denoising, we have taken an image which has 512*512 of spatial resolution, and followed the procedure curated in Saragadam et al. (2023), Photon noise was simulated in our experiment by assigning a Poisson random variable, distributed independently at each pixel, with a maximum mean photon count of 30 and a readout count of 2, which results a noise image with PSNR and SSIM of 15.49, 0.173 respectively, and this can be considered as an extremely noisy situation with the size of the image. Then, INRs were tested on the ability of image denoising as can be seen from figure 14, PIN is the only architecture which was able to recover the PSNR to 27.35 dB, which signifies an improvement of nearly 12 dB. This substantial increase in PSNR reflects a significant reduction in noise and restoration of image quality. Additionally, PIN achieved the highest SSIM value of 0.78, illustrating its effectiveness not only in reducing noise but also in maintaining the structural integrity and perceptual quality of the image. On the other hand, the ReLU+PE combination emerged as the second most effective activation for denoising extremely noisy images, demonstrating the second most robust performance.

### A.5.3 PIN AS A RELAIBLE EDGE DETECTOR

INRs represent signals implicitly, and a crucial aspect of a good INR is its capability to perform tasks associated with explicit representations. In image processing, explicit representations allow us to extract edges using filters like Prewitt, Sobel, or Gaussian, vital for tasks such as object boundary identification, text detection, or facial feature recognition, which heavily depend on accurate edge

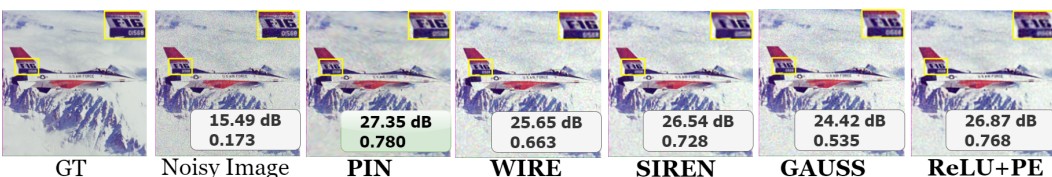

Figure 14: **Image denoising capabilities of PIN**: In this task, INRs were assigned to denoise images initially characterized by a PSNR of 15.49dB and SSIM of 0.173. The recovered images demonstrate the denoising capabilities of INRs. PIN's superior metrics show its proficiency in restoring images with minimal distortion while preserving maximum structural information.

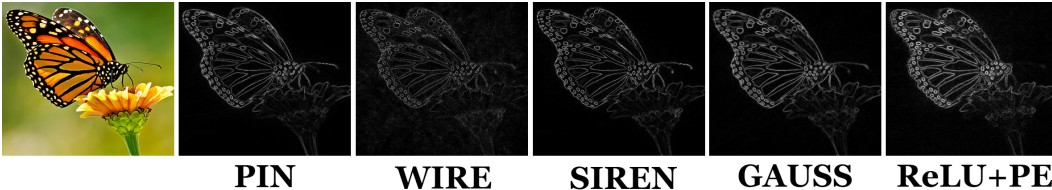

**PIN**      **WIRE**      **SIREN**      **GAUSS**      **ReLU+PE**

Figure 15: **PIN as an edge detector**: The results above showcase the edge detection capabilities of INRs. Among the various models evaluated, PIN emerges as a standout performer, demonstrating its ability to produce the cleanest and most well-defined edge maps with minimal false edge detections.

detection. While theoretically, we could derive edge maps from an INR via its decoded representations, INRs offer the advantage of differentiable learned representations. Therefore, a superior INR should not only represent the signal accurately but also provide access to its gradient information. For an INR to effectively act as an edge detector, it should encode pixel-level correspondences in its weights and biases without regularization. The potential of identifying edge maps through INRs becomes evident when presented with images containing visually identifiable edges, such as the butterfly image depicted on the top left of figure 15. Initially, the image is represented using INRs, and thereafter once the representation is encoded into the INR, the gradient operator is applied between the learned representation and the training coordinates. figure 15 depicts the butterfly's RGB image alongside its corresponding edge maps obtained through INRs. PIN stands out by providing the cleanest and most well-defined edge map, identifying necessary edges with minimal false edge identifications. Among other INRs, GAUSS, despite producing smooth edge maps, sacrifices intricate details, which can diminish the true edge signal and result in a less reliable representation. Even though WIRE, and ReLU+PE showcase the boundaries, as an edge detector they fail significantly due to noise enhancement effects. Therefore, this demonstrate that even though these INRs are capable of doing the representation, when it comes to down stream tasks like accessing gradients through the encoded representation, their weights and biases do not adequtely maintain the necessary pixel level correspondances. On the other hand, SIREN also functions well as an edge detector, benefiting from its specific weight initialization, specific frequency parameter, and inherent edge detection capabilities. Even without such inherent features, PIN accurately identifies edges, emphasizing its ability to retain intricate pixel-level correspondences without any regularization schemes while providing accurate representations. Thus, PIN stands out as a reliable edge detector compared to existing state-of-the-art INRs.

