# OpenReview forum: "PIN: Prolate Spheroidal Wave Function-based Implicit Neural Representations"
_ICLR.cc/2025/Conference — ICLR 2025 Poster_

### Official Review · Reviewer_BT7N · 2024-10-17

**Soundness:** 3
**Presentation:** 2
**Contribution:** 3
**Rating:** 6
**Confidence:** 4

**Summary:**

This paper argues that the Gaussian and Gabor wavelet activation functions cannot achieve the optimal space-frequency trade-off, and thus may not effectively capture distant relevance. Motivated by this, the paper proposes using the Legendre polynomial numerical estimation of the Prolate Spheroidal Wave Function (PSWF) as the activation function, which has been proven in previous work to offer the optimal space-frequency trade-off. The authors demonstrate improvements through experiments on various tasks, including image regression, neural radiance fields, and so on.

**Strengths:**

1. The motivation presented in the paper is compelling and provides a solid foundation for the proposed approach.
2. This paper offers a insightful critique of previous works, identifying limitations in existing activation functions and highlighting the need for improvement.
3. The experiments encompass a diverse range of tasks, allowing for a more comprehensive and thorough comparison. This diversity enhances the validity of the findings and demonstrates the effectiveness of the proposed method across different domains.

**Weaknesses:**

1. The exact implementation of the numerical estimation of the Prolate Spheroidal Wave Function lacks clarity in the main text. For instance, the approximation order is not explicitly stated, and the impact of the approximation order on the computational complexity (in terms of space/time consumption) and the model's performance is not clearly explained.
2. The experiments in tasks, such as Occupancy Field Representation and Neural Radiance Field, appear to be conducted on a subset of the entire dataset, which may undermine the persuasiveness of the results.
3. The presentation could be improved by refining certain details, such as employing the vector graphics and employing the \autoref{} command for figure citations to ensure consistency and clarity.

**Questions:**

1. Could you clarify the approximation order used for the Prolate Spheroidal Wave Function (PSWF)? Additionally, does this approximation affect the theoretical properties of the PSWF? The paper mentions that PSWF possesses infinite support in space, but it remains unclear whether the finite-order approximation may potentially diminish this property.
2. I observed that a bandwidth parameter \(c\) governs the frequency. Could you provide guidance on how to select this parameter effectively? Does it influence model performance, and if so, what strategies should users follow to determine an optimal value, especially for tasks in continuous domains like Neural Radiance Fields (NeRF), where there might be no frequency bounds, unlike the image regression scenario?

---

> ### Author Response · Authors · 2024-11-21
>
> First and foremost, the authors would like to thank the reviewer for their thoughtful and insightful questions, which have provided us with an excellent opportunity to clarify and strengthen our work.
>
> W1:
>
> We sincerely thank the reviewer for their insightful question. We acknowledge that the numerical implementation has not been detailed in the main text and will make sure to incorporate this in the revised version. We would also like to gently point out that a more comprehensive description is available in Appendix regarding the approximation method we used. Specifically, we utilized cubic spline approximation, which provides a third-order approximation between two successive points while maintaining continuity and differentiability. A simple regression would often fail to retain the exact data point, and the differentiability properties between discrete points, where differentiability properties are indeed needed during the backpropagation. Therefore, the cubic spline approximation, which is of third-order and computationally efficient, serves as an optimal choice, and does the intended task.
>
> W2:
> We are thankful for the reviewer regarding his question. The following table summarizes the occupancy field results for different INRs.
>
>
> **Table: IoU for Occupancy Fields**
>
> | **Model**     | **Asian Dragon** | **Armadillo** | **Happy Buddha** | **Lucy**  |
> |---------------|------------------|---------------|-------------------|-----------|
> | Siren         | 0.95473          | 0.97685       | 0.98155           | 0.96503   |
> | Wire          | 0.93780          | 0.95674       | 0.95618           | 0.99797   |
> | Gauss         | 0.99620          | 0.98919       | 0.99594           | 0.99060   |
> | ReLU+PE       | 0.98362          | 0.99274       | 0.99824           | 0.98211   |
> | PIN           | 0.99837          | 0.99824       | 0.99895           | 0.99917   |
>
> The following table summarizes NeRF results
>
> **Table: NeRF for different objects**
>
> | **Object**     | **PIN**  | **WIRE** | **SIREN** | **GAUSS** |
> |----------------|----------|----------|-----------|-----------|
> | Chair          | 33.590   | 30.328   | 31.610    | 33.233    |
> | Hotdog         | 36.340   | 33.452   | 31.261    | 36.224    |
> | Mic            | 33.150   | 29.060   | 31.725    | 32.761    |
> | Ship           | 28.822   | 25.902   | 27.252    | 28.767    |
> | Materials      | 29.654   | 26.184   | 28.028    | 29.407    |
> | Ficus          | 27.235   | 22.592   | 24.556    | 26.940    |
>
> W3:
> We are thankful for the suggestion by the reviewer. The authors will make sure to improve the presentation of the paper by utilizing vector graphics and the suggested commands in the revised manuscript.
>
> Q1:
> We utilized a cubic spline approximator, which is order 3. The authors do not believe this would affect any theoretical properties of the PSWFs, as when we get the discretized solution to the governing equation for PSWF, we utilized a cubic spline approximator between every successive points. This cubic spline approximation is necessary because a continuous function, rather than a set of discrete points, is required as an activation function in a neural network. So as far as our knowledge is concerned there is no any potential way of diminishing any theoretical properties. However, if a simpler approximator such as a basic quadratic or cubic polynomial, or even a neural network, were used to approximate the discretized solution, it could potentially compromise some of the properties of PSWFs. These methods do not guarantee passing through the discretized points, which could lead to inaccuracies in differentiation and, consequently, incorrect outcomes during backpropagation.
>
> Q2:
>
> We are grateful for reviewer regarding this question. As explained in Section 6 of the paper, we utilize an explicit control mechanism, where the frequency is now governed by the parameter \( \omega \). This makes \( \omega \) the frequency-controlling variable instead of \( c \). Now, regarding how to effectively select the parameter \( \omega \), one possible approach is to perform a grid search to identify the values that maximize the results. However, as you mentioned, this often leads to suboptimal outcomes when a different dataset is presented, particularly in NeRF and other applications where strong generalization is crucial. (Incidentally, this is the standard approach adopted by many INR baselines.)  Instead of relying on a grid search, we use the most straightforward configuration for \( \omega \), which is \( \omega = 1 \). We initialize all experiments with this value and make \( \omega \) a learnable parameter. Consequently, it gets adjusted dynamically based on the loss function of the intended task. This approach benefits from our spline approximation, which allows the activation function parameters to traverse in the loss landscape more effectively, as explained in the Appendix of the paper

---

> > ### Author Response · Authors · 2024-11-24
> >
> > We sincerely believe we have addressed the reviewer's questions, including the additional results and detailed explanations for the approximation method. As the deadline for the discussion period approaches (November 26th), we wanted to gently follow up to see if you've had the opportunity to review our response. If there are any further questions or clarifications needed, please let us know. We would be more than happy to provide detailed answers to ensure all concerns are fully addressed.

---

> > > ### Comment · Reviewer_BT7N · 2024-11-24
> > >
> > > Thank you for your response, and I sincerely apologize for the delayed reply. The authors have addressed all my questions, and I would like to maintain my original voting score.

---

> > > > ### Author Response · Authors · 2024-11-28
> > > >
> > > > We sincerely thank the reviewer for the thoughtful and insightful questions, and these provided us with an excellent opportunity to clarify and further strengthen our work.

---

### Official Review · Reviewer_1n58 · 2024-10-22

**Soundness:** 2
**Presentation:** 2
**Contribution:** 3
**Rating:** 6
**Confidence:** 3

**Summary:**

This paper introduces the use of prolate spheroidal wave functions (PSWF) for implicit neural representation (INR). By employing PSWF as the activation function in INR, the method excels not only in representing images and 3D shapes but also significantly outperforms existing approaches in various vision tasks that rely on INR generalization, including image inpainting, novel view synthesis, edge detection, and image denoising.

**Strengths:**

1. Extensive experiments across various vision tasks demonstrate the effectiveness of PSWF.

2. A comprehensive theoretical analysis highlights the advantages of using PSWF.

**Weaknesses:**

1. When comparing different INR methods, do you ensure that the same parameters are used? Could you provide the specific parameters for each INR method?

2. I am curious about the decoding complexity of the PSWF-based INR. Could you provide the decoding speed or time for the different INR methods?

3. Besides the vision tasks mentioned in the paper, can PSWF also improve performance in image super-resolution?

**Questions:**

I noticed that the authors use initial INR methods as their baselines. However, there are several approaches aimed at enhancing the expressivity and generalizability of INR, including improvements in training strategies [1] and input signals [2]. Could PSWF be applied to these methods to further enhance the representation performance of INR?


[1] Improved Implicit Neural Representation with Fourier Reparameterized Training, CVPR 2024

[2] Disorder-invariant implicit neural representation, CVPR 2023.

**Details Of Ethics Concerns:**

There are no ethics concerns.

---

> ### Author Response · Authors · 2024-11-21
>
> First and foremost, the authors would like to thank the reviewer for their thoughtful and insightful questions, which have provided us with an excellent opportunity to clarify and strengthen our work.
>
> W1:
>
> For all the experiments, we have ensured the optimal parameters of the other methods have been used. However, for the proposed method, every experiment has been conducted with the same parameters unlike others. As can be seen, these baselines do require specific fine-tuning to get the results. However, PIN does not require any of those conditions.
> The following table summarizes the activation function parameters utilized for each application.
>
> **Table: Configurations for WIRE, SIREN, GAUSS, and PIN Across Experiments**
>
> | **Method** | **Configuration**                                                                                                                                      |
> |------------|--------------------------------------------------------------------------------------------------------------------------------------------------------|
> | WIRE       | Image Representation, Inpainting, Edge Detection ($\omega=20$, $\sigma=10$), Occupancy Field ($\omega=20$, $\sigma=40$), Image Denoising ($\omega=5$, $\sigma=5$), NeRF ($\omega=40$, $\sigma=40$) |
> | SIREN      | $\omega=30$ for all experiments                                                                                                                       |
> | GAUSS      | $\sigma=30$ for all except NeRF ($\sigma=7.85$)                                                                                                        |
> | PIN        | $T=1$, $\omega=1$, $b=0$ for all                                                                                                                       |
>
> W2:
>
> We are thankful for the reviewer regarding the question. The following table summarizes the training speed corresponding to different INRs.  As can be seen from this table, PIN's runtime is comparable with that of previous INRs.
>
> **Table: Convergence Time Across Methods**
>
> | **Method**    | **Convergence Time (min)** |
> |---------------|-----------------------------|
> | PIN           | 6.63                        |
> | WIRE          | 11.59                       |
> | SIREN         | 6.29                        |
> | GAUSS         | 7.47                        |
> | ReLU+PE       | 4.05                        |
>
> W3:
>
> We are thankful to the reviewer for raising this question. We attempted the image super-resolution task on the "Boy" image from the Set14 dataset [ref] and the "Cameraman" image. The following table summarize the results for the "Boy" and "Cameraman" images in 2nd and 3rd columns respectively.
>
> **Table: PSNR for Image Super Resolution**
>
> | **Method**    | **PSNR (dB)** | **PSNR (dB)** |
> |---------------|---------------|---------------|
> | PIN           | 20.97         | 23.73         |
> | WIRE          | 19.26         | 22.33         |
> | SIREN         | 19.58         | 23.17         |
> | GAUSS         | 20.24         | 22.70         |
> | ReLU+PE       | 18.75         | 21.67         |
>
> [ref]. Awesome-Super-Resolution/dataset.md at master ·559
> ChaofWang/Awesome-Super-Resolution — github.com.560
> https : / / github . com / ChaofWang / Awesome -561
> Super-Resolution/blob/master/dataset.md.562
> [Accessed 20-11-2024]
>
> Q1:
>
> We are thankful for the question, and the suggestions. The authors acknowledge that PIN is compared with initial methods. When considering improvements to [1], we believe there is potential to enhance its methods using PSWFs. However, the core idea of [1] is to employ a fixed Fourier basis and decompose the neural network weight matrix into a product of trainable and fixed Fourier basis matrices. Given this, a key question arises: how can the Fourier basis and PSWFs be effectively combined? A possible approach is to modify [1] by incorporating Fourier basis elements of PSWFs, and using the same weight update rule as in [1].However, without experimental validation, it is difficult to definitively state whether this adaptation would further enhance [1].
> When it comes to enhancing [2], which basically looks the problem in another way, more specifically the input; we firmly believe PSWFs can be incorporated into [2], as their proposal mechanism is based on the input. Detailed experiments are indeed needed to verify the claim. To further demonstrate the effectiveness of the proposed method, we compared the proposed approach with recently released FINER, INCODE, FR-INR on the entire Kodak image dataset. The following table summarizes the results. The suggested references, new methods, and additional results will be included in the revised version of the paper.
>
> **Table: Comparison of Methods Based on PSNR (dB)**
>
> | **Method**    | **PSNR (dB)** |
> |---------------|---------------|
> | PIN           | 40.17         |
> | INCODE        | 34.07         |
> | FR-INR        | 37.91         |
> | FINER         | 35.87         |

---

> > ### Comment · Reviewer_1n58 · 2024-11-22
> >
> > Many thanks to the author for addressing my concerns. I have no further questions. I keep my current score unchanged.

---

> > > ### Author Response · Authors · 2024-11-28
> > >
> > > We sincerely thank the reviewer for the thoughtful and insightful questions, and these provided us with an excellent opportunity to clarify and further strengthen our work.

---

### Official Review · Reviewer_Yext · 2024-11-02

**Soundness:** 3
**Presentation:** 3
**Contribution:** 3
**Rating:** 6
**Confidence:** 5

**Summary:**

The paper proposes a novel Implicit Neural Representation (INR) utilizing Prolate Spheroidal Wave Functions (PSWFs) to improve performance and generalization in computer vision tasks. By leveraging the optimal space-frequency domain concentration of PSWFs, the proposed method addresses the noise artifacts over smoother areas and poor generalization of existing INRs, demonstrating superior results in image inpainting, novel view synthesis, edge detection, and image denoising.

**Strengths:**

1. The paper clearly explains the limitations of current INRs and proposes a novel activation function (PSWFs) for INR to overcome these limitations.
2. The localization and expressivity properties of PIN have been theortically proven.
3. The results validate the effectiveness of the proposed INR across various tasks. The authors not only show the good representation ability of PIN for various signals, such as image, Occupancy Filed, and NeRF, but also show that PIN has a very good performance on image Image Inpainting.

**Weaknesses:**

1. The activation function seems to be rather computationally heavy, making it necessary to report its speed in applications.
2. As I have reviewed the previous submission of this paper, I notice that the Fig.4 is replaced with a scene with better results. I wonder how these new results are obtained and why these results are not attached in previous submission? Are they obtained by tuning parameters for each scene specifically?
3. Since the first NeurIPS submission of this paper, several new INRs have been proposed, including FINER (and its extension, FINER++) and H-SIREN. However, these new INRs are not cited or compared within the current manuscript.

**Questions:**

See Weakness.

---

> ### Author Response · Authors · 2024-11-21
>
> First and foremost, the authors sincerely thank the reviewer for their thoughtful and insightful questions. Furthermore, we greatly appreciate your previous comments, which have been instrumental in refining our work and providing us with an excellent opportunity to further clarify and strengthen it.
>
> W1:
>
> The effective training times for the proposed method is as follows.  As can be seen from the table, PIN's runtime is comparable with that of previous INRs.
>
> **Table: Convergence Time Across Methods**
>
> | **Method**    | **Convergence Time (min)** |
> |---------------|-----------------------------|
> | PIN           | 6.63                        |
> | WIRE          | 11.59                       |
> | SIREN         | 6.29                        |
> | GAUSS         | 7.47                        |
> | ReLU+PE       | 4.05                        |
>
> W2:
>
> We thank the reviewer for their thoughtful question and careful review. We would like to clarify that the improved results were not obtained by fine-tuning our proposed method on specific datasets or scenes. Unlike existing baselines, our method uses the same configuration across all experiments, regardless of the data modality.
>
> In the previous submission, we reported results for a 3D dataset, but it did not significantly contrast our results with other INRs. During the rebuttal for the previous submission, we noted that PIN performs well on all 3D datasets except the one used in that submission. Therefore, for this submission, we replaced the previously reported results with those from different 3D datasets to better showcase the performance gap of our method.
>
> W3:
>
> We are thankful for the reviewer for the question, and the suggestions. The authors acknowledge that we compared PIN with initial methods. However, to demonstrate the effectiveness of the proposed method, we compared the proposed approach with recently released FINER, INCODE, FR-INR on the entire Kodak image dataset. The following table summarizes the results. Further, the authors will make sure to cite the suggested and latest methods in the revised version of the manuscript.
>
> **Table: Comparison of Methods Based on PSNR (dB)**
>
> | **Method**    | **PSNR (dB)** |
> |---------------|---------------|
> | PIN           | 40.17         |
> | INCODE        | 34.07         |
> | FR-INR        | 37.91         |
> | FINER         | 35.87         |

---

> > ### Comment · Reviewer_Yext · 2024-11-22
> >
> > Could you please provide a comparison about the network size in the Tab.2 of your response?

---

> > > ### Author Response · Authors · 2024-11-22
> > >
> > > We are thankful for the reviewer regarding the question. For a fair comparison, we utilized an MLP with 300 hidden neurons and 5 layers for all methods.

---

> > > > ### Comment · Reviewer_Yext · 2024-11-24
> > > >
> > > > Thank you. All of my concerns have been addressed. I reserve my original voting score.

---

> > > > > ### Author Response · Authors · 2024-11-28
> > > > >
> > > > > We sincerely thank the reviewer for the thoughtful and insightful questions, and these provided us with an excellent opportunity to clarify and further strengthen our work.

---

### Official Review · Reviewer_8yGC · 2024-11-03

**Soundness:** 3
**Presentation:** 3
**Contribution:** 2
**Rating:** 6
**Confidence:** 2

**Summary:**

The paper proposes Prolate Spheroidal Wave Function-based Implicit Neural Representations (PIN), an effective representation inspired by Prolate Spheroidal Wave Functions (PSWFs). The proposed PIN outperforms other INR baselines in various reconstruction tasks.

**Strengths:**

- The proposed PIN outperforms other INR baselines in various reconstruction tasks, including image reconstruction, image inpainting, and occupancy field and NeRF.
- Detailed ablation studies are conducted.

**Weaknesses:**

- The metrics provided in the paper are only evaluated on a few individual examples. I appreciate the evaluation on the Kodak Lossless
True Color Image Dataset. However, it only contains 24 images. It is ideal to perform larger-scale evaluations, e.g. calculating the mean PSNR for a thousand images. For example, one may consider using the DIV2K dataset [1].

[1] Agustsson E, Timofte R. Ntire 2017 challenge on single image super-resolution: Dataset and study[C]//Proceedings of the IEEE conference on computer vision and pattern recognition workshops. 2017: 126-135.

**Questions:**

- I am interested in the runtime of the proposed PIN. Will the new formulation hurt the speed of INRs? Is there a trade-off between quality and runtime?
- How is the experiment conducted for the Image Inpainting task? Are the missing pixels marked as black and fed into the INR? Or are those coordinates masked out and not used during training? Are the pixel masks provided to the INR?

---

> ### Author Response · Authors · 2024-11-21
>
> First and foremost, the authors would like to thank the reviewer for their thoughtful and insightful questions, which have provided us with an excellent opportunity to clarify and strengthen our work.
>
> W1:
> We sincerely thank the reviewer for emphasizing the importance of larger-scale evaluations and for recognizing our use of the Kodak Lossless True Color Image Dataset. Evaluating INRs necessitates retraining the model for each image, making large-scale assessments computationally demanding. While many major baselines in INR research evaluate their methods on only two or three examples, we are, to the best of our knowledge, the first to conduct a comprehensive evaluation across the entire Kodak dataset. In addition, we extended our analysis to the DIV2K dataset as suggested, even though with some limitations. Due to time and resource constraints, we evaluated PIN and other state-of-the-art (SOTA) methods on a randomly selected subset of 30 images from DIV2K. The average PSNR values for this subset are presented in the table1 below, and as shown, PIN outperforms other methods on this subset as well. Given that these images were selected randomly, we believe this performance pattern is representative of PIN’s general superiority and would likely extend to the entire DIV2K dataset. Combining these results, we report the overall average PSNR metrics for 54 images (24 from Kodak and 30 from DIV2K) in table2. We deeply value the reviewer's suggestion regarding larger-scale evaluations and are actively considering this for future work. Specifically, leveraging meta-learning or other training efficiency mechanisms could make such evaluations more feasible. Thank you again for highlighting this aspect and for your constructive feedback. Further, we will incorporate these results to the revised manuscript.
>
> **Table 1: PSNR Variation across DIV2K dataset**
>
> | **Method**    | **PSNR (dB)** |
> |---------------|---------------|
> | PIN           | 41.46         |
> | WIRE          | 30.12         |
> | SIREN         | 38.77         |
> | GAUSS         | 28.13         |
> | ReLU+PE       | 26.74         |
>
>
> **Table 2: PSNR Variation Across DIV2K and Kodak Datasets**
>
> | **Method**    | **PSNR (dB) Avg of (DIV2K + KODAK)** |
> |---------------|-------------------------------------|
> | PIN           | 40.88                              |
> | WIRE          | 31.61                              |
> | SIREN         | 38.05                              |
> | GAUSS         | 27.19                              |
> | ReLU+PE       | 27.48                              |
>
>
> Q1:
>
> We are thankful for the reviewer's question. We computed run-times for the convergence. The following table provides the run-times.  As can be seen from the table, PIN's runtime is comparable with that of previous INRs.
>
> **Table: Convergence Time Across Methods**
>
> | **Method**    | **Convergence Time (min)** |
> |---------------|-----------------------------|
> | PIN           | 6.63                        |
> | WIRE          | 11.59                       |
> | SIREN         | 6.29                        |
> | GAUSS         | 7.47                        |
> | ReLU+PE       | 4.05                        |
>
> Q2:
>
> We are thankful for the reviewer regarding the question. When it comes to INRs, they are based on the coordinates of the signal that is being provided to it. So for training the inpainting task, the coordinates corresponding to the inpainted regions are masked out, and during the testing time the entire coordinates of the image is provided. This will effectively asses the method's generalization abilities for unseen coordinates.

---

> > ### Author Response · Authors · 2024-11-24
> >
> > We sincerely believe we have addressed the reviewer's questions, including a thorough evaluation on the DIV2K dataset and detailed explanations for the inpainting task. As the deadline for the discussion period approaches (November 26th), we wanted to gently follow up to see if you've had the opportunity to review our response. If there are any further questions or clarifications needed, please let us know. We would be more than happy to provide detailed answers to ensure all concerns are fully addressed.

---

> > > ### Author Response · Authors · 2024-11-25
> > >
> > > Dear Reviewer 8yGC,
> > >
> > > We hope this message finds you well. As the discussion period is set to conclude tomorrow, and noting that the other reviewers have already responded, we wanted to kindly follow up to check if you have had the chance to review our response. If there are any additional questions or areas requiring clarification, please let us know. We would be happy to provide detailed answers to ensure all your concerns are fully addressed.
> > >
> > > Thank you for your time and consideration.

---

> > > > ### Comment · Reviewer_8yGC · 2024-11-27
> > > >
> > > > Dear authors,
> > > >
> > > > Thanks for the detailed response and the additional evaluations. My concerns are well resolved and I am happy to raise the score.

---

> > > > > ### Author Response · Authors · 2024-11-28
> > > > >
> > > > > We sincerely thank the reviewer for the thoughtful and insightful questions, and these provided us with an excellent opportunity to clarify and further strengthen our work.

---

### Comment · Area_Chair_YjYe · 2024-11-27
**Please take a look at the authors' rebuttal and start a discusssion if needed**

Dear Reviewers,

Thanks for your contributions in reviewing this paper.

As the author-reviewer discussion deadline is approaching, please could you take a look at the authors' rebuttal (if not yet) and see if it addressed your concerns or if you have any further questions. Please feel free to start a discussion.

Thanks,

AC

---

### Meta-Review · Area_Chair_YjYe · 2024-12-21

**Metareview:**

In this paper, the authors presented a new activation function for implicit neural representations (INRs) -- Prolate Spheroidal Wave function (PSWF). Motivated by the challenges faced by existing INRs and their struggle to generalise to unseen coordinates, the authors introduced the PSWF-based INRs, termed PIN, leveraging the optimal space-frequency domain concentration of PSWFs. Experimental evaluations over a few different vision tasks (including image inpainting, novel view synthesis, edge detection, and image denoising) show the effectiveness of the proposed PIN. The strengths of this paper include:
- The proposed method was well-motivated, with a clear and solid foundation for the approach.
- The paper did a good job in analysing and explaining the limitations of existing INRs, which could provide insights for following research in this direction.
- The proposed method was backed up with a comprehensive theoretical analysis.
- An extensive experimental analysis covering several vision tasks, showing the effectiveness and validity of the proposed method.

The weaknesses of this paper include:
- The rationale for employing INRs to address low-level problems (as raised by a reviewer during 2nd phase discussion, details see below). Forward training-based models typically offer better generalisation capabilities.
- Potential issues with the infinitely differentiable property of cubic spline that was used for the PSWF in this paper (also confirmed in the 2nd phase).
- Insufficient evaluations, computational complexity concern, issues with the results, and missing comparison to recent related works.

Most of the concerns/weaknesses were well addressed in the rebuttal phase and the reviewers also acknowledged that. Overall, this paper presented an interesting idea with insights into INRs, and the AC think this would be of interest to a group of audience in ICLR. As a result, the AC is happy to recommend an Accept, but the authors are **highly suggested to incorporate the further provided evidence and clarifications during the discussions to the final version, and please also merge the Appendix (currently in a separate file), which has some essential analysis and results, to the end of the main paper.**

**Additional Comments On Reviewer Discussion:**

This paper received review comments from four expert reviewers. During the rebuttal period, there was a heated discussion between the authors and reviewers. with the additionally provided results and evidence by the authors, most of the concerns raised by the reviewers were well addressed, and two reviewers raised their ratings, ending up with 4 borderline Accept. In the AC-reviewers discussion phase, reviewer BT7N further summarised the strengths and weaknesses of this paper, including the concerns about the rationale for applying INRs to low-level problems and the differentiable property of cubic spline. The AC agreed with them, while found them not major issues. However, the authors are suggested to carefully consider these points and add discussions in their final version.

Although this paper finally received a borderline rating, after carefully checking the paper and the discussions, the AC found this paper could provide insights to the community, and a group of audience in ICLR can benefit from reading it, as a result, worth being presented at ICLR. These led to the final decision of this paper.

---

### Decision · Program_Chairs · 2025-01-22

Accept (Poster)